# Clip-plate versus suture-anchor in double-door laminoplasty for degenerative cervical myelopathy: Protocol for a multicenter, non-inferiority, randomized controlled trial

Kentaro Yamada[1], Kenichiro Sakai[2], Takashi Hirai[1], Kazuyuki Fukushima[3], Takuya Takahashi[2], Yu Matsukura[1], Satoru Egawa[1], Hiroaki Onuma[1], Motonori Hashimoto[1], Akihiro Hirakawa[4], Yoshiyasu Arai[2], Toshitaka Yoshii ![ORCID][1]*

1 Department of Orthopaedics, Institute of Science Tokyo, Tokyo, Japan, 2 Department of Orthopaedics, Saiseikai Kawaguchi Sogo Hospital, Saitama, Japan, 3 Department of Orthopaedics, Saku Medical Center, Saku, Japan, 4 Department of Clinical Biostatistics, Institute of Science Tokyo, Tokyo, Japan

* yoshii.orth@tmd.ac.jp

## Abstract

### Background

Degenerative cervical myelopathy (DCM) is a leading cause of spinal cord dysfunction. While cervical laminoplasty (LAMP) is the standard treatment, the optimal fixation method for the opened lamina, specifically between the conventional suture-anchor method and the modern plate system, remains debated. Although clip-plates offer rigid fixation for the opened lamina, they are associated with higher costs and potential complications such as lamina reclosure due to screw dislodgement or hinge fracture. This study aims to verify the non-inferiority of the suture-anchor method compared to the clip-plate method in double-door LAMP.

### Methods

This is a multicenter, open-label, non-inferiority randomized controlled trial. A total of 216 patients with DCM (cervical spondylotic myelopathy [CSM] or ossification of the posterior longitudinal ligament [OPLL]) will be recruited from three high-volume spine centers in Japan. Participants will be randomized (1:1) to undergo double-door LAMP using either clip-plates (control group) or suture-anchors (experimental treatment group). The primary endpoint is the recovery rate of the Japanese Orthopaedic Association (JOA) score at 1 year postoperatively. Secondary endpoints include operative time, blood loss, clinical assessments (EQ-5D-5L, visual analog scale [VAS], and neck disability index [NDI]), radiological outcomes (cervical spinal alignment, lamina retention rate, hinge fracture, bone union, and MRI findings), direct medical costs, and perioperative complications.

which permits unrestricted use, distribution, and reproduction in any medium, provided the original author and source are credited.

**Data availability statement:** This is a protocol paper, and data have not yet been generated. The datasets that will be generated during the current study will not be publicly available due to ethical restrictions imposed by the Institutional Review Board (IRB) of Institute of Science Tokyo. Any secondary use of the data will require individual approval from the IRB to protect patient privacy, as specified in the protocol (S2 and S3 Files). Data access requests should be directed to the corresponding author or the IRB of Institute of Science Tokyo(contact email: rinri.adm@tmd.ac.jp).

**Funding:** This research was supported by the grant of AO Spine Asia Pacific Research National Grant (AOSRG2024078). The funder had no role in study design, data collection and analysis, decision to publish, or preparation of the manuscript.

**Competing interests:** All authors declared that no competing interests exist.

## Conclusions

We hypothesize that the suture-anchor technique is non-inferior to the clip-plate system regarding neurological recovery and laminar stability. By rigorously comparing these techniques, this trial seeks to establish high-level evidence for a surgical strategy that maintains clinical standards while potentially reducing healthcare costs.

## Trail registration

Japan Registry of Clinical Trials (jRCT) jRCT1032250437

## Introduction

Degenerative cervical myelopathy (DCM) encompassing Cervical spondylotic myelopathy (CSM) including degenerative disc disease and ossification of the posterior longitudinal ligament (OPLL) are common degenerative spinal disorders that can lead to significant neurological impairment and disability [1]. Cervical laminoplasty (LAMP), a posterior decompression technique designed to preserve the posterior elements of the cervical spine was developed in Japan to reduce problems frequently associated with laminectomy, such as postoperative spinal instability, kyphotic deformity, and formation of the postlaminectomy membrane [2]. LAMP is currently regarded as the gold standard surgical treatment for DCM, particularly for patients presenting without kyphotic deformity or anterior spinal cord compression [3–5]. The surgical procedure of LAMP is generally categorized into two techniques: double-door [6] and open-door LAMP [2]. The clinical outcomes between the two techniques are reported to the comparative [7–9].

A complication specific to cervical LAMP is "lamina closure" which can lead to unfavorable long-term outcomes including cervical kyphotic change and neurological deterioration [10–13]. To prevent lamina closure, recreating a rigid posterior neural arch to obtain bony fusion at the hinge or split spinous process is essential. In conventional double-door LAMP described by Kirita and Miyazaki [6], the laminae were maintained in an open position using stay sutures threaded through the spinous process/the yellow ligament and facet capsule/paravertebral muscle. Subsequently, various materials were introduced to maintain lamina expansion, including hydroxyapatite spacers, suture-anchors, or mini-plates [14].

Suture anchors represent a relatively novel devices for this purpose in the double-door LAMP, offering a simplified technique that requires only embedding the anchor into the lateral mass, thereby avoiding the complex and time-consuming placement of bulkier implants [15,16]. Biomechanical study demonstrated that the use of the suture anchor in cervical LAMP was sufficiently biomechanically tolerable [17]. A pilot study utilizing intraoperative CT demonstrated that the suture-anchor technique maintained laminar expansion more rigidly than the traditional suture-only method in double-door LAMP despite small sample size [18].

Conversely, mini-plate systems were developed in double-door LAMP to provide a more rigid and durable fixation and are currently widely utilized [19–21]. Recently,

clip-type plate implants have been reported to exhibit 1.5- 2.0-fold increase in biomechanical strength compared to conventional hydroxyapatite spacers [22]. However, lamina closure can still occur even in the mini-plate system with the incidence of 3.4–13.4% by screw dislodgement or hinge fracture [19,20]. Because the mini-plate systems offer additional increased instrumentation costs, the major concern was whether the mini-plate system provide superior merit than the other alternative techniques.

A recent cost-effectiveness study comparing the mini-plate and hydroxyapatite spacers demonstrated that the surgical costs and incremental cost-effectiveness ratio (ICER) were higher for mini-plates, despite no difference in clinical and radiological outcomes [23]. Although meta-analyses suggested similar clinical results between with and without plate technique [24], high-quality randomized controlled trials (RCTs) comparing clinical outcomes, safety profiles, and cost-effectiveness are lacking. Consequently, the true merit of the more expensive plate technique remains unverified.

We hypothesized that the conventional suture-anchor technique yields clinical and radiological outcomes that are non-inferior to the clip-plate technique in the double-door LAMP. The objective of this study is to conduct a multicenter, non-inferiority, RCT to compare the clinical efficacy, radiological outcomes, and safety of double-door laminoplasty using suture-anchors versus clip-plate systems. The study also aims to demonstrate that the conventional technique is non-inferior to the clip-plate procedures, potentially offering a more cost-effective alternative without compromising patient outcomes.

## Methods

### Trial design

This is a multicenter, open-label, non-inferiority randomized controlled trial. All trial procedures are summarized in Fig 1. Patients will be randomized in a 1:1 ratio to one of two groups: the control group (double-door LAMP using clip-plates) or the experimental treatment group (double-door LAMP using suture-anchors), (Fig 2).

### Ethical approval and registration

This study protocol was approved by the Institute of Science Tokyo Certified Review Board (I2024−243). This study will be conducted in strict adherence to the Declaration of Helsinki and the Clinical Trials Act of Japan. This trial was registered at Japan Registry of Clinical Trials (jRCT: jRCT1032250437) on October 17, 2025.

### Study population

A total of 216 patients scheduled for surgical treatment of LAMP for an MRI/CT-confirmed cervical DCM (CSM or OPLL) will be recruited from the Department of Orthopaedic Surgery at one university hospital and two high-volume spine centers located in Japan. The enrollment period is from October 17, 2025 to September 30, 2028. Written informed consent will be obtained from all participants. Data collection will be completed by September 30, 2030. Results will be analyzed by March 31, 2031. Patients must meet all the following criteria to be eligible to participate.

**Inclusion criteria.**

1) Patients with spinal cord symptoms due to DCM (CSM or OPLL) scheduled for cervical LAMP involving the expansion of 2–4 laminae at C3-6.

2) Patients with confirmed spinal cord stenosis at the C3-7 level on MRI or CT

3) Patients aged 20 years or older but under 90 years at the time of consent

4) Patients capable of providing written informed consent

| | Study Period | | | | | | | |
|---|---|---|---|---|---|---|---|---|
| | Enrollment | Allocation | Surg. | Post-allocation | | | Close-out | |
| TIMEPOINT *(Acceptable time window)* | *Preop* | | 0 | *PO.1w* | *Dis.* | *PO.30d (±14d)* | *PO. 1Y (±56d)* | *PO. 2Y (±56d)* |
| **ENROLLMENT:** | | | | | | | | |
| **Eligibility screen** | X | | | | | | | |
| **Informed consent** | X | | | | | | | |
| **Randomization** | | X | | | | | | |
| **INTERVENTIONS:** | | | | | | | | |
| *[Experimental Treatment]: Suture-anchor* | | | X | | | | | |
| *[Control]: clip-plate* | | | X | | | | | |
| **ASSESSMENTS:** | | | | | | | | |
| *Patient Backgrounds* | X | | | | | | | |
| *Medical History, Physical Examination* | X | | | X | X | X | X | X |
| *Surgical Information* | | | X | | | | | |
| *Clinical assessment* (JOA score, EQ5D, VAS, NDI) | X | | | | | | X | X |
| *Cervical X-ray* (cervical spinal alignment) | X | | | X | | | X | X |
| *Cervical Plain CT* (Retention rate, Hinge fracture, Bone union) | X | | | X | | | X | X |
| *Cervical Plain MRI* (paraspinal muscle CSA, Dural sac CSA, Grading of the mass posterior to the dural sac) | X | | | | | | X | X |
| *Direct Medical Costs* | | | ←————————————————→ | | | | | |
| *Intraoperative complications/Adverse events* | | | X | | | | | |
| *Early postoperative complications/Adverse events* | | | ←——————————→ | | | | | |
| *Late postoperative complications/Adverse events* | | | | | | ←——————→ | | |

**Fig 1. SPIRIT schedule of enrolment, interventions, and assessments.** The schedule outlines the timeline for participant enrollment, interventions, and assessments in the study. SPIRIT, Standard Protocol Items: Recommendations for Interventional Trials; Surg., Surgery; PO., postoperative; Dis., discharge; JOA, Japanese orthopaedic association; NDI, Neck disability index; VAS, Visual analog scale; CSA, Cross-sectional area.

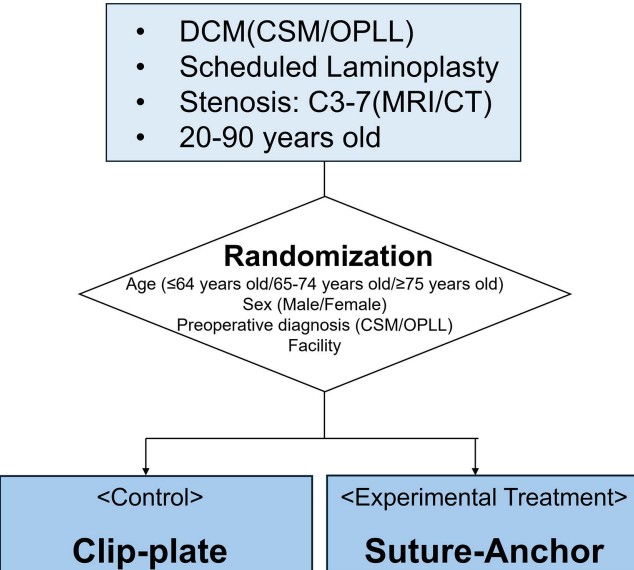

**Fig 2. Study flowchart.** DCM, Degenerative cervical myelopathy; CSM, Cervical spondylotic myelopathy; OPLL, Ossification of the posterior longitudinal ligament.

**Exclusion criteria.**

1) Inappropriate surgical indication: Patients with significant cervical kyphosis, anterior spinal cord compression, or cervical spinal instability on MRI or CT, where LAMP is contraindicated due to the risk of insufficient decompression or progression of deformity.

2) Concomitant foraminal stenosis: Patients requiring concomitant posterior foraminal enlargement (foraminotomy), as this may confound operative time and outcome assessments.

3) Active spinal infection

4) Spinal tumors: including metastatic tumors

5) Traumatic spinal cord injury

6) History of prior cervical spine surgery

7) Maintenance dialysis: Due to the high prevalence of destructive spondyloarthropathy and poor bone quality, which may affect implant fixation and fusion.

8) Cerebral palsy: Involuntary movements can adversely affect postoperative alignment.

9) Parkinson's disease: high muscle tone in these conditions can adversely affect postoperative alignment or functional recovery scores.

10) Pregnancy: to avoid radiation exposure from protocol-mandated CT scans.

11) Other: Any patient judged unsuitable for the study by the attending surgeon.

## Randomization

Eligible patients will be randomized using a minimization method to ensure balance between groups. Stratification factors include age (≤64 years / 65–74 years / ≥75 years), sex, preoperative diagnosis (CSM vs. OPLL), and study facility. The randomization process will be managed via a secure electronic data capture (EDC) system.

## Blinding

Due to the distinct nature of the implants visible on postoperative imaging, blinding of surgeons and patients is not feasible. However, data analysts will be blinded to group allocation where possible during the statistical analysis phase.

## Treatment

All surgeries must be supervised by a Japanese Society of Spine Surgery and Related Research (JSSR) certified spine surgery instructor.

**Control group: clip-plate technique.** Double-door LAMP using clip-plates is to be conducted as follows. A midline incision and exposure of the C3–C6 laminae, and the spinous processes are trimmed. The laminae are split at the center using a high-speed drill, and bilateral gutters are created at the laminofacet junction. The laminae are gently opened. Small pilot holes are drilled through the opened laminae, and mini-screws are inserted to secure a polyether ether ketone clip-plate (LAMINAclip2, Olympus Terumo Biomaterial Corp., Tokyo, Japan) across the gap by placing over the screwheads. Although this clip-type plate implant differs from traditional plate systems, it is classified and biomechanically validated as a spinal plate in Japan [22]. This maneuver is performed for at least two levels (Fig 3).

**Experimental treatment group: Suture-anchor technique.** The surgical exposure and laminar splitting are identical to the control group. After opening the bilateral split laminae gently, self-tapping titanium suture anchors (LAMIFIX, Olympus Terumo Biomaterial Corp., Tokyo, Japan) are inserted into the lateral masses, taking care not to penetrate the facet joints. Sutures attached to the anchors are threaded through the yellow ligament and tied to maintain the laminae in an open position. If the stability of the non-fixed lamina is inadequate, alternatively, the additional suture is made between the yellow ligament and the deep fascia. This maneuver is performed for at least two levels (Fig 4).

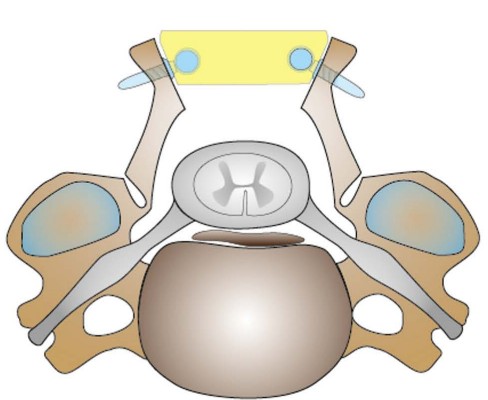 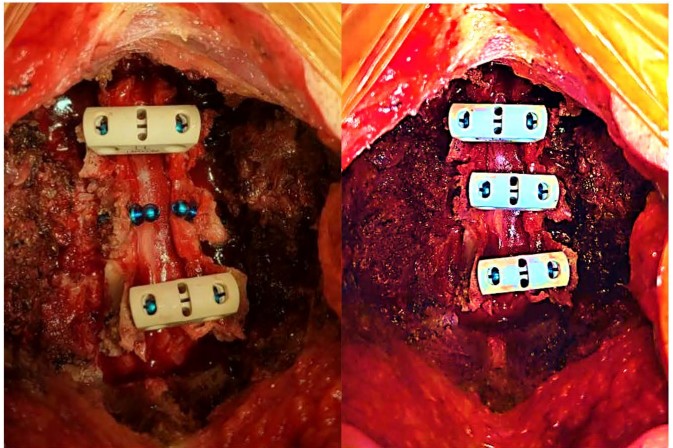

**Fig 3. Schema and intraoperative photographs of double-door LAMP using clip-plates.** Mini-screws are set through the opened laminae. Then, the clip-plate is placed over the screw head. LAMP, laminoplasty.

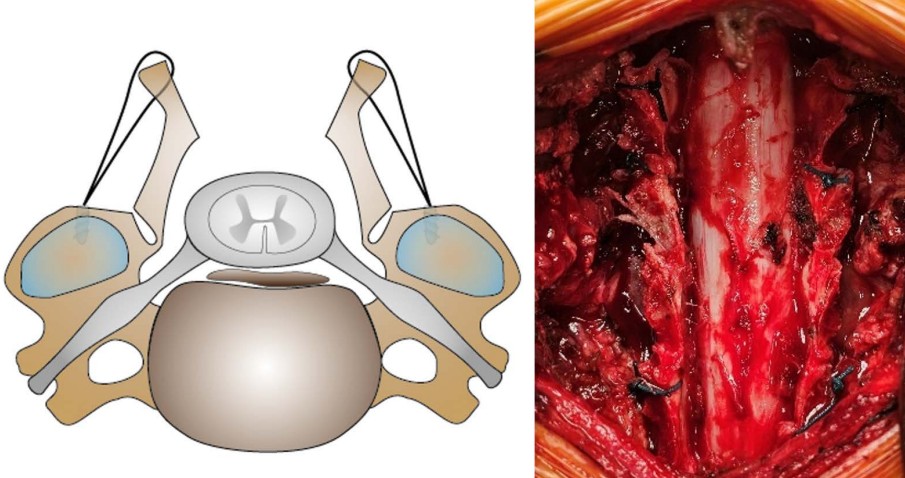

**Fig 4. Schema and intraoperative photographs of double-door LAMP using suture-anchors.** Suture-anchors are inserted into the lateral mass. Then, the sutures are threaded through the yellow ligament and tied in the open position of lamina. LAMP, laminoplasty.

## Perioperative management

Postoperative management, including antibiotics, analgesia, and rehabilitation, will follow the standard of care at each facility. The use of a cervical collar is not mandated but must be recorded if used. The timing of postoperative ambulation and the content of rehabilitation are not specified, but rehabilitation must be performed for all patients.

## Follow-up and data collection

After surgical treatment, the patients will attend orthopaedic outpatient visits every one to six months. At each visit, the attending surgeon will assess the adverse event, including revision surgery. If any adverse event occurs, the attending surgeon must report it using the case report form.

The clinical outcomes are to be measured at preoperatively, postoperatively 1 and 2 years by validated self-reported questionnaires and by physical examination. The radiological outcomes include X-ray, CT and MRI examination at preoperatively, 1 and 2 years postoperatively in addition to postoperative 1-week CT. The schedule of enrollment, interventions, and assessments is shown in Fig 1.

To ensure data quality, a clinical research coordinator will enter data from medical records, and central monitoring will be implemented. To safeguard patients' personal information, each patient will be assigned a unique identification code. All data will be stored as password-protected electronic files on the EDC system with access restricted to investigators only.

## Endpoints

***Primary endpoint.*** The Primary endpoint is the recovery rate of cervical Japanese Orthopaedic Association (JOA) score (Table 1) at 1 year postoperatively. The recovery rate calculation formula is as previously reported [2] as follows.

*Cervical JOA Score recovery rate* = (*Postoperative Score − Preoperative Score*) / (17 − *Preoperative Score*) × 100 (%)

The primary analysis time point is postoperative 1 year. Results at postoperative 2 years are also presented as a final analysis.

**Table 1. Cervical JOA score. Maximum total score:17.**

| Motor function (8 points) | |
|---|---|
| **Upper extremity (4 points)** | |
| 0 | Complete function loss |
| 1 | Possible to eat with spoon, but not with chopsticks and impossible to write |
| 2 | Possible to eat with chopsticks or to to write but inadequate |
| 3 | Possible to eat with chopsticks or to write, awkward |
| 4 | Normal |
| **Function of Schoulder-elbow (−2 points)** | |
| −2 | Strength of biceps brachii or deltoid ≤ Grade 2 |
| −1 | Strength of biceps brachii or deltoid = Grade 3 |
| −0.5 | Strength of biceps brachii or deltoid = Grade 4 |
| 0 | Strength of biceps brachii or deltoid = Grade 5 |
| **Lower extremity (4 points)** | |
| 0 | Impossible to stand and walk |
| 0.5 | Possible to stand, impossible to walk |
| 1 | Needs cane or aid to walk on flat ground |
| 1.5 | Possible to walk independently on flat ground, awkward |
| 2 | Need cane or aid on stairs |
| 2.5 | Need cane or aid on downward stairs only |
| 3 | Possible to walk without cane or aid, but slowly |
| 4 | Normal |
| **Sensory function (6 points)** | |
| **Upper extremity (2 points), trunk (2 points), and lower extremity (2 points)** | |
| 0 | Complete sensory loss |
| 0.5 | Apparent disturbanace, less than 5/10 sensory, unbearable pain or numbness |
| 1 | Moderate disturbance, more than 6/10 sensory, moderatre numbness, hypersensitivity |
| 1.5 | Mild disturbance, mild numbness, normal touch |
| 2 | Normal |
| **Bladder function (3points)** | |
| 0 | Complete retention |
| 1 | Severe disturbance, sense of retention, dribbling, imcomplete continence |
| 2 | Mild disturbance, urinary frequency, urinary hesitancy |
| 3 | Normal |

JOA, Japanese orthopaedic association

### Secondary endpoint.

(1) Operative Time and estimated blood loss

(2) Minimal Clinically Important Difference (MCID) achievement: the proportion of patients achieving MCID for Cervical JOA score recovery rate, defined as 52.8% according to previous reports [25], at postoperative 1 and 2 years postoperatively in each group.

(3) Patient-reported outcomes: health-related quality of life (EQ-5D-5L), Visual Analogue Scale (VAS) for neck pain, upper limb pain, and upper limb numbness, Neck disability index (NDI) at postoperative 1 and 2 years

(4) Radiological outcomes:

- Retention rate: The ratio of the laminar angle at postoperative 1 or 2 years to the angle at 1 week postoperative on CT. A rate < 0.8 indicates lamina closure [13].

- Hinge fracture: Graded as 0 (none), 1 (ventral cortex fracture), or 2 (complete fracture with collapse) on CT [16] at postoperative 1 or 2 years.

- Bone union of gutter: Classified as Union (bony bridging), Healed (callus without bridging), or Nonunion on CT [16] at postoperative 1 or 2 years.

- Cervical spinal alignment: C2-7 angle including the range of motion calculated from flexion and extension lateral views, C-SVA, and T1 slope on X-ray at postoperative 1 and 2 years.

- Paravertebral muscle: Paraspinal muscle cross-sectional area (CSA) in C4/5 level at postoperative 1 and 2 years on MRI.

- Postlaminectomy Membrane: Grading of the soft tissue mass posterior to the dural sac on the central slice of the T2 sagittal image [26] and dural sac CSA (C3/4, C4/5, C5/6, C6/7 levels) at postoperative 1 and 2 years.

(5) Health Economics: evaluated by ICER. Costs will include direct medical costs (surgery, hospitalization, management of adverse events). Quality-Adjusted Life Years (QALYs) will be derived from EQ-5D-5L scores using a Markov model, with costs and outcomes discounted at 2% annually.

(6) Adverse Events: Incidence of intraoperative, early (<30 days), and late (>30 days) complications, using the evaluation criteria from the JSSR database [27] and grading by the Common Terminology Criteria for Adverse Events V.5.0. [28]

## Sample size calculation

Based on a preliminary unpublished study, the mean ± standard deviation JOA recovery rate was 49.9 ± 22.8% for the plate group and 52.6 ± 32.9% for the suture-anchor group. The non-inferiority margin was set at 20%. As no previous non-inferiority RCTs have compared these techniques, this margin was determined by expert consensus to be approximately half of the known MCID (52.8% [25]) for the JOA recovery rate, ensuring a clinically acceptable difference regardless of potential benefits in operative time or cost.

Assuming a control recovery rate of 50% ± 40% and a non-inferiority margin of 20%, with a one-sided alpha of 0.025 and 90% power, 172 subjects are required. Accounting for 20% attrition, the target sample size is 216 (108 per group).

## Data monitoring

An independent committee will monitor the data annually. Any event reasonably suspected to have a causal relationship with the study intervention will be considered an adverse event and promptly reported to the investigators for evaluation. Adverse events will be thoroughly recorded, tracked, and reported to the Ethics Committee in accordance with requirements regarding severity, causality, and expectedness. Follow-up will continue until the events are resolved or stabilized. The Principal Investigator will conduct a cumulative review of all adverse events and convene an investigator meeting when necessary to assess the study's risks and benefits. No formal audits are planned for this trial.

## Statistical analysis

Efficacy analysis will be performed on the Full Analysis Set (FAS). The FAS is defined as all randomized participants who underwent the protocol treatment (cervical laminoplasty), excluding those with no post-randomization data. The primary

endpoint (JOA recovery rate at 1 year) will be analyzed using analysis of covariance, adjusting for age, sex, preoperative diagnosis, and baseline JOA score. Non-inferiority is established if the lower limit of the 95% confidence interval for the difference in least squares means (Suture-anchor minus Clip-plate) exceeds −20%. Secondary endpoints will be compared using t-tests, Wilcoxon rank-sum tests, or logistic regression to appropriately account for the stratification variables when comparing the binary outcomes between randomized groups. Longitudinal data will be analyzed using mixed-effects models. Safety analyses will be conducted on the Safety Analysis Set (SAS). The SAS is defined as all randomized participants who underwent the protocol treatment. Specifically, any individual lamina with an intraoperative hinge fracture clearly identified by the attending surgeon will be treated as a "non-responder" within the FAS for the analysis of the lamina retention rate, hinge fracture, and bone union of the gutter.

## Discussion

DCM, including CSM and OPLL, is the most common cause of spinal cord dysfunction in the elderly worldwide. The incidence and prevalence of DCM are estimated at a minimum of 41 and 605 per million in North America. The overall incidence of CSM-related hospitalizations has been estimated at 4.04 per 100,000 person-years, and surgical rates seem to be rising [1]. As the aging population grows, the incidence of DCM-related hospitalizations continues to rise, necessitating effective and sustainable surgical interventions [29], While cervical laminoplasty (LAMP) has become the gold standard for multisegmental DCM, avoiding the kyphotic deformities and membrane formation associated with laminectomy [30], the optimal fixation method for the opened lamina remains a subject of debate.

The double-door LAMP technique proposed by Kirita and Miyazaki has been accepted as an effective compression myelopathy treatment [3,31,32]. While effective, this method has the potential to cause "lamina reclosure," a complication occurring in 1.8–5.5% of cases that is associated with neurological deterioration [13,16]. The plate system was initially established as an alternative technique to open-door LAMP, providing rigid fixation and simplifying the surgical workflow [33]. It has also been developed in the technique of double-door LAMP and widely spread nowadays [19–22]. However, this mechanical advantage is not without drawbacks; screw back-out and dislodgement, potentially lead to failure of retaining opened lamina [19,20]. Furthermore, the significant cost differential between plates and sutures, with the latter being substantially more expensive, raises critical questions regarding value-based healthcare [29].

This study represents the first large-scale randomized controlled trial (RCT) designed to directly compare the modern clip-plate technique against the conventional suture-anchor technique in double-door LAMP. The primary hypothesis is that the suture-anchor method is non-inferior to the clip-plate system in terms of neurological recovery, as measured by the JOA score. By validating non-inferiority, this trial aims to challenge the assumption that more expensive hardware is associated with superior clinical outcomes. The inclusion of specific exclusion criteria, such as Parkinson's disease and maintenance dialysis, ensures that the analysis focuses purely on the efficacy of the fixation method for the opened lamina without the confounding variables of severe comorbidities affecting motor function.

Beyond clinical efficacy, this trial addresses several distinct gaps in the current literature. First, safety profiles regarding hinge fractures and bone union will clarify whether the rigid fixation of plates is truly safer than the semi-rigid fixation of anchors. Specifically, the incidence of hinge fractures, a common complication that can lead to palsy or closure, will be meticulously tracked via CT grading [16]. Second, the radiological assessment of the "postlaminectomy membrane" using MRI offers novel insights. Postlaminectomy membrane was reported to occur even after double-door laminoplasty using hydroxyapatite spacers [34]. The actual incidence of postlaminectomy membrane remains unclear after LAMP because no cohort studies have investigated whether LAMP prevents this membrane. Combining analysis of the grading of the mass posterior to the dural sac and cross-sectional area between the Kirita-Miyazaki LAMP with suture-anchors, which involves LAMP without a midline bony roof, and LAMP using clip-plates to form the lamina roof, will yield new insights regarding the incidence of postlaminectomy membrane after LAMP. Finally, the health economic analysis will use ICER. In an era of escalating healthcare costs, demonstrating that a lower-cost technique yields equivalent outcomes could justify changes

in insurance reimbursement and surgical guidelines. Japanese insurance data indicates a four-fold cost difference between the implants, highlighting the potential for significant systemic savings [23].

Furthermore, the study employs a minimization randomization method stratified by age, sex, diagnosis, and facility. This rigorous design choice addresses the heterogeneity inherent in DCM populations, ensuring that the treatment groups are balanced regarding critical prognostic factors.

In conclusion, this prospective RCT anticipates establishing high-level evidence regarding the comparative utility of suture-anchors versus clip-plates. By balancing clinical efficacy, safety, and economic sustainability, the findings will empower surgeons and policymakers to make evidence-based decisions in the management of DCM. If the conventional suture-anchor technique proves non-inferior, it validates a cost-effective alternative that maintains high standards of patient care, effectively bridging the gap between traditional surgical principles and modern economic realities.

## Supporting information

**S1 File. SPIRIT checklist.**
(DOCX)

**S2 File. Copy of the protocol that was approved by the ethics committee (Japanese).**
(DOCX)

**S3 File. Copy of the protocol that was approved by the ethics committee (English).**
(DOCX)

## Acknowledgments

We would like to thank the members of Health Science Research & Development Center (HeRD) at the Institute of Science Tokyo for their assistance in preparing protocol of this clinical research. **AI Assistance Statement:** Gemini 2.5 pro (Google, CA, USA) was utilized for English proofreading and translation assistance. The authors retain full responsibility for the content.

## Author contributions

**Conceptualization:** Toshitaka Yoshii.

**Data curation:** Kentaro Yamada, Kenichiro Sakai, Kazuyuki Fukushima, Takuya Takahashi, Satoru Egawa, Yoshiyasu Arai, Toshitaka Yoshii.

**Funding acquisition:** Toshitaka Yoshii.

**Methodology:** Akihiro Hirakawa.

**Writing – original draft:** Kentaro Yamada.

**Writing – review & editing:** Kenichiro Sakai, Takashi Hirai, Kazuyuki Fukushima, Takuya Takahashi, Yu Matsukura, Satoru Egawa, Hiroaki Onuma, Motonori Hashimoto, Akihiro Hirakawa, Yoshiyasu Arai, Toshitaka Yoshii.

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
