## [Decision Letter · Decision Letter 0]

2 Jan 2026

Dear Dr. Yoshii,

Thank you for submitting your manuscript to PLOS ONE. After careful consideration, we feel that it has merit but does not fully meet PLOS ONE’s publication criteria as it currently stands. Therefore, we invite you to submit a revised version of the manuscript that addresses the points raised during the review process.

We look forward to receiving your revised manuscript.

Kind regards,

Koji Akeda

Academic Editor

PLOS One

Journal Requirements:

Additional Editor Comments:

The manuscript entitled ‘Mini-plate versus Suture-Anchor in Double-door Laminoplasty for Degenerative Cervical Myelopathy: Protocol for a Multicenter, Non-Inferiority, Randomized Controlled Trial’ aims to assess the non-inferiority of the suture-anchor method compared to the mini-plate method in the point of clinical outcomes for the primary endpoint. This study also evaluates the radiological outcomes, perioperative complications and medical costs for secondary outcomes.

Thank you for submitting this well-designed study protocol, which explores a clinically relevant question regarding mini-plate applications in cervical spine surgery. The reviewers have provided constructive feedback to strengthen the manuscript; please address their comments in your revision.

Reviewer's Responses to Questions

**Comments to the Author**

1. Does the manuscript provide a valid rationale for the proposed study, with clearly identified and justified research questions?

Reviewer #1: Yes

Reviewer #2: Yes

Reviewer #3: Yes

2. Is the protocol technically sound and planned in a manner that will lead to a meaningful outcome and allow testing the stated hypotheses?

Reviewer #1: Yes

Reviewer #2: Yes

Reviewer #3: Partly

3. Is the methodology feasible and described in sufficient detail to allow the work to be replicable?

Reviewer #1: Yes

Reviewer #2: Yes

Reviewer #3: Yes

4. Have the authors described where all data underlying the findings will be made available when the study is complete?

The PLOS Data policy requires authors to make all data underlying the findings described in their manuscript fully available without restriction, with rare exception, at the time of publication. The data should be provided as part of the manuscript or its supporting information, or deposited to a public repository. For example, in addition to summary statistics, the data points behind means, medians and variance measures should be available. If there are restrictions on publicly sharing data—e.g. participant privacy or use of data from a third party—those must be specified.requires authors to make all data underlying the findings described in their manuscript fully available without restriction, with rare exception, at the time of publication. The data should be provided as part of the manuscript or its supporting information, or deposited to a public repository. For example, in addition to summary statistics, the data points behind means, medians and variance measures should be available. If there are restrictions on publicly sharing data—e.g. participant privacy or use of data from a third party—those must be specified.

Reviewer #1: No

Reviewer #2: No

Reviewer #3: Yes

5. Is the manuscript presented in an intelligible fashion and written in standard English?

Reviewer #1: Yes

Reviewer #2: Yes

Reviewer #3: Yes

You may also provide optional suggestions and comments to authors that they might find helpful in planning their study.

Reviewer #1: I appreciate the opportunity to review this interesting protocol manuscript. DCM is a major societal problem, and since surgical treatment is often unavoidable, re-evaluating optimized treatment strategies from an objective perspective is critically important. The authors have designed the RCT appropriately and are investing considerable effort to evaluate numerous clinically important outcomes. This effort is commendable, and I eagerly await the results.

That said, some portions warrant improvement or clarification. I would appreciate the authors' thoughts on the points outlined below.

1.

Upon reviewing this RCT, I cannot definitively state that the title is accurate. Is this device (LAMINAclip2) truly a "mini-plate"? Judging from the photos, it appears more like a block than a plate. Furthermore, it seems designed to be inserted into a protrusion of the screw embedded in the lamina, rather than being fixed to the lamina with screws. To me, the mechanism of this device seemed closer to a threaded HA spacer than a plate-screw system.

Plate-screw systems derive their fixation force from the static friction generated by the plate being pressed tightly against the bone by the torque of the screws. Alternatively, in a locking plate system, the device achieves mechanical properties similar to external fixation through the locking of the plate and screws. Does the LAMINAclip2 possess such mechanical characteristics?

While this is my personal opinion, I believe the title creates a gap between what readers might expect and the actual content.

Therefore, how about considering the following?: Provide evidence demonstrating that this device possesses the same biomechanical characteristics as the plate-screw system, replace "mini-plate" with an appropriate term, or simply use a mini-plate system in the control group.

2.

I recommend including cervical range of motion (ROM) in pre- and postoperative radiographic evaluations. This is because the importance of cervical ROM has gained increasing attention in recent years when exploring risk factors for postoperative cervical kyphosis.

- Fujishiro T, Obo T, Yamamoto Y, et al. J Clin Neurosci 2024

- Ren H, Shen X, Ding R, et al. Spine. 2023

3.

The term "fixation" is used several times in the text. I cannot dismiss the concern that using this term alone might lead readers to misunderstand this paper as discussing "cervical fusion surgery." Therefore, please ensure it is made absolutely clear that the paper describes a fixation method specifically for the opened lamina. The phrase should be written as "the fixation method for the opened lamina," not simply "the fixation method."

Once again, I wish to state that the authors' efforts are commendable. I am very much looking forward to the results.

Reviewer #2: Thank you for a well presented study.

I have some minor clarifications.

Abstract: consider re-phrasing "and even exist potential complications"

line 142 and 262 and 314: gender - is it gender or sex? Fig 2 indicates it is male/ female - use sex rather than gender, and indicate it is sex at birth somewhere if this is what is intended.

line 261: what is the definition for the FAS?

line 267: what is the definition for the SAS?

line 266: analysing for differences in randomised group by chi-squared test will not account for the stratification variables - consider logistic regression. Please specify that all analyses by randomised group will appropriately consider the stratification variables.

line 281: "is associated with" rather than "correlates"

Reviewer #3: This manuscript presents a well-designed protocol for a multicenter randomized controlled trial comparing suture-anchor–based and mini-plate–based double-door laminoplasty for degenerative cervical myelopathy. The study addresses a clinically relevant question , and the overall study design, endpoints, and statistical analysis plan are appropriate. I have the following two minor comments.

1. Management of intraoperative hinge fractures In double-door laminoplasty, intraoperative hinge fractures can occur, which may make it difficult to evaluate bone union. The protocol should clarify how these cases will be handled. Specifically, please state whether these cases will remain eligible for evaluation or be treated as protocol deviations. Please also clarify how they will be analyzed (e.g., intention-to-treat).

2. Rationale for the non-inferiority margin According to the CONSORT extension for noninferiority trials, the rationale for the non-inferiority margin should be stated. The current protocol uses a 20% margin based on the MCID. However, further clinical rationale for selecting "20%" would be helpful. For example, please explain if this 20% margin is consistent with previous studies, or if it is an acceptable difference considering the potential benefits of the suture-anchor technique (such as shorter operative time or lower cost).

.

Reviewer #1: No

Reviewer #2: No

Reviewer #3: No

---

## [Author Response · Author response to Decision Letter 1]

20 Feb 2026

Dear Editor

Additional Editor Comments:

The manuscript entitled ‘Mini-plate versus Suture-Anchor in Double-door Laminoplasty for Degenerative Cervical Myelopathy: Protocol for a Multicenter, Non-Inferiority, Randomized Controlled Trial’ aims to assess the non-inferiority of the suture-anchor method compared to the mini-plate method in the point of clinical outcomes for the primary endpoint. This study also evaluates the radiological outcomes, perioperative complications and medical costs for secondary outcomes.

Thank you for submitting this well-designed study protocol, which explores a clinically relevant question regarding mini-plate applications in cervical spine surgery. The reviewers have provided constructive feedback to strengthen the manuscript; please address their comments in your revision.

Response: Thank you very much for your positive feedback and for the opportunity to revise our manuscript. We appreciate your recognition of the clinical relevance of our study.

We have carefully reviewed all the comments provided by the reviewers and have addressed each of them in the revised manuscript. We believe that these revisions have significantly improved the quality and clarity of our protocol.

Dear Reviewers,

The detailed review of this manuscript is greatly appreciated, and your comments have been helpful in allowing us to revise our manuscript. We have re-checked our study protocol according to the reviewers’ comments. We have addressed the questions raised by the reviewers. The revised sentences in the manuscript have been highlighted with red color.

Reviewer #1:

I appreciate the opportunity to review this interesting protocol manuscript. DCM is a major societal problem, and since surgical treatment is often unavoidable, re-evaluating optimized treatment strategies from an objective perspective is critically important. The authors have designed the RCT appropriately and are investing considerable effort to evaluate numerous clinically important outcomes. This effort is commendable, and I eagerly await the results.

That said, some portions warrant improvement or clarification. I would appreciate the authors' thoughts on the points outlined below.

Response: Thank you for your encouraging comments and for recognizing the importance of our study. We appreciate your positive evaluation of our RCT design and its clinical significance.

In response to your valuable feedback, we have carefully addressed the points requiring improvement and clarification. Our detailed responses to each of your suggestions are provided below.

1. Upon reviewing this RCT, I cannot definitively state that the title is accurate. Is this device (LAMINAclip2) truly a "mini-plate"? Judging from the photos, it appears more like a block than a plate. Furthermore, it seems designed to be inserted into a protrusion of the screw embedded in the lamina, rather than being fixed to the lamina with screws. To me, the mechanism of this device seemed closer to a threaded HA spacer than a plate-screw system.

Plate-screw systems derive their fixation force from the static friction generated by the plate being pressed tightly against the bone by the torque of the screws. Alternatively, in a locking plate system, the device achieves mechanical properties similar to external fixation through the locking of the plate and screws. Does the LAMINAclip2 possess such mechanical characteristics?

While this is my personal opinion, I believe the title creates a gap between what readers might expect and the actual content.

Therefore, how about considering the following?: Provide evidence demonstrating that this device possesses the same biomechanical characteristics as the plate-screw system, replace "mini-plate" with an appropriate term, or simply use a mini-plate system in the control group.

Response: Thank you for your insightful comment regarding the classification of the device. We apologize for the insufficient explanation of the control group’s implant in the original manuscript.

As you pointed out, the LAMINAclip2 differs from the traditional screw-and-plate mechanism.

To more accurately reflect its unique mechanism and clinical function, we have adopted the term "clip-plate" throughout the revised manuscript including the title. This device is a PEEK clip-type plate implant fixed by snapping it onto the screw heads.

To address your concerns regarding its mechanical properties, we have added a reference to a recent biomechanical study (reference # 22).This study demonstrated that these implants exhibited a 1.5- to 1.7-fold higher reaction force than spacers at the middle of the lamina, and a 1.9- to 2.0-fold higher reaction force at the tip of the lamina compared to traditional HA spacers.

Based on these biomechanical results, this device is officially approved and reimbursed under the category of "064 Spinal Fixation Materials (2) Spinal Plate (Standard Type)" by the Japanese insurance system, rather than as a spacer, as shown in the Table below.

Therefore, the manufacturer refer to this implant as “clip plate” as following the product catalog.

We have added a description regarding the mechanical strength of the clip-plate in the Methods section, along with the relevant supporting literature.

P4 Line 65-67: Recently, clip-type plate implants have been reported to exhibit 1.5- 2.0-fold increase in biomechanical strength compared to conventional hydroxyapatite spacers [22].

P8, Line 161-162: Although this clip-type plate implant differs from traditional plate systems, it is classified and biomechanically validated as a spinal plate in Japan [22].

Reference [22]: Mui T, Kawasaki S, Shigematsu H, Ikejiri M, Sada T, Sinthubua A, et al. Biomechanical Evaluations of Novel Clip-Type Implants for Cervical Double-Door Laminoplasty, Compared with Conventional Hydroxyapatite Spacers: A Cadaveric Study. S Spine Surg Relat Res. 2026;10(1):73-9. doi: 10.22603/ssrr.2025-0192.

2. I recommend including cervical range of motion (ROM) in pre- and postoperative radiographic evaluations. This is because the importance of cervical ROM has gained increasing attention in recent years when exploring risk factors for postoperative cervical kyphosis.

- Fujishiro T, Obo T, Yamamoto Y, et al. J Clin Neurosci 2024

- Ren H, Shen X, Ding R, et al. Spine. 2023

Response: Thank you for this important suggestion. We fully agree with the significance of assessing cervical range of motion (ROM) in relation to postoperative kyphosis, as highlighted in the literature you provided.

In our original protocol, we planned to perform preoperative and postoperative lateral radiographs in both flexion and extension positions, in addition to the neutral position. This allows for the measurement of the C2/7 angle in each position to evaluate ROM. To make this clearer, we have added a description of ROM evaluation to the "Radiological outcomes" section in the revised manuscript.

P12, Line231-232: Cervical spinal alignment: C2-7 angle including the range of motion calculated from flexion and extension lateral views, C-SVA, and T1 slope on X-ray at postoperative 1 and 2 years.

3. The term "fixation" is used several times in the text. I cannot dismiss the concern that using this term alone might lead readers to misunderstand this paper as discussing "cervical fusion surgery." Therefore, please ensure it is made absolutely clear that the paper describes a fixation method specifically for the opened lamina. The phrase should be written as "the fixation method for the opened lamina," not simply "the fixation method."

Once again, I wish to state that the authors' efforts are commendable. I am very much looking forward to the results.

Response: We appreciate your careful reading and the suggestion to clarify our terminology. We understand that the term "fixation" alone could be misinterpreted as "spinal fusion surgery" by some readers.

In accordance with your advice, we have revised the manuscript to ensure that the term is more specific. We have changed "fixation method" to "fixation method for the opened lamina" in several locations, including the Abstract and the Discussion section as follows, to clearly indicate that our study focuses on the stabilization of the expanded laminae in laminoplasty.

P1, Line 8-9: While cervical laminoplasty (LAMP) is the standard treatment, the optimal fixation method for the opened lamina, specifically between the conventional suture-anchor method and the modern plate system, remains debated.

P1, Line10: Although clip-plates offer rigid fixation for the opened lamina,

P15, Line 291-292: the optimal fixation method for the opened lamina remains a subject of debate.

P16, Line 308-311: The inclusion of specific exclusion criteria, such as Parkinson’s disease and maintenance dialysis, ensures that the analysis focuses purely on the efficacy of the fixation method for the opened lamina without the confounding variables of severe comorbidities affecting motor function.

Reviewer #2:

Thank you for a well presented study.

I have some minor clarifications.

Response: Thank you very much for your thoughtful and constructive comments. We have revised the manuscript to address each of your points as follows:

Abstract: consider re-phrasing "and even exist potential complications"

Response: We appreciate your suggestion to clarify the phrasing in the Abstract. We have revised the sentence to improve the flow and grammatical accuracy as below. We believe this more clearly describes the relationship between the costs and specific clinical risks.

P1, Line 10-13: Although clip-plates offer rigid fixation for the opened lamina, they are associated with higher costs and potential complications such as lamina reclosure due to screw dislodgement or hinge fracture. This study aims to verify the non-inferiority of the suture-anchor method compared to the clip-plate method in double-door LAMP.

line 142 and 262 and 314: gender - is it gender or sex? Fig 2 indicates it is male/ female - use sex rather than gender, and indicate it is sex at birth somewhere if this is what is intended.

Response: Following your suggestion, we have replaced "gender" with "sex" throughout the manuscript; Line 143, 271, 328, and Fig 2. We have also clarified that this refers to sex at birth.

line 261: what is the definition for the FAS?

line 267: what is the definition for the SAS?

Response: In accordance with our study protocol, we have added specific definitions for the FAS and SAS in the Statistical Analysis section as follows.

P14, Line 268-270: The FAS is defined as all randomized participants who underwent the protocol treatment (cervical laminoplasty), excluding those with no post-randomization data.

P14, Line 277-278: The SAS is defined as all randomized participants who underwent the protocol treatment.

line 266: analysing for differences in randomised group by chi-squared test will not account for the stratification variables - consider logistic regression. Please specify that all analyses by randomised group will appropriately consider the stratification variables.

Response: We appreciate the advice regarding the stratification variables. We have revised the text to specify that all analyses will appropriately account for these variables using logistic regression instead of the chi-squared test where applicable.

P14, Line 273-276: Secondary endpoints will be compared using t-tests, Wilcoxon rank-sum tests, or logistic regression to appropriately account for the stratification variables when comparing the binary outcomes between randomized groups.

line 281: "is associated with" rather than "correlates"

Response: We have changed "correlates" to "is associated with" as your suggestion.

P15 Line 295-296: a complication occurring in 1.8–5.5% of cases that is associated with neurological deterioration

Reviewer #3:

This manuscript presents a well-designed protocol for a multicenter randomized controlled trial comparing suture-anchor–based and mini-plate–based double-door laminoplasty for degenerative cervical myelopathy. The study addresses a clinically relevant question , and the overall study design, endpoints, and statistical analysis plan are appropriate. I have the following two minor comments.

Response: We are grateful for your positive evaluation of our study protocol and for recognizing its clinical relevance. We also appreciate your comments regarding the appropriateness of our study design and statistical analysis plan. In response to your comments, we have updated the manuscript to provide further clarification

1. Management of intraoperative hinge fractures In double-door laminoplasty, intraoperative hinge fractures can occur, which may make it difficult to evaluate bone union. The protocol should clarify how these cases will be handled. Specifically, please state whether these cases will remain eligible for evaluation or be treated as protocol deviations. Please also clarify how they will be analyzed (e.g., intention-to-treat).

Response: Thank you for this valuable comment. We have clarified the management of intraoperative hinge fractures in the revised manuscript.

First, all efficacy analyses will be conducted using the Full Analysis Set (FAS) based on the Intention-to-Treat (ITT) principle. Specifically, any individual lamina with an intraoperative hinge fracture clearly identified by the attending surgeon will be treated as a "non-responder" within the FAS for the analysis of the postoperative lamina retention rate, hinge fracture, and bone union of the gutter. We believe this approach minimizes potential bias by including all randomized participants while ensuring a conservative and robust evaluation of the surgical outcomes.

P14, Line 278-281: Specifically, any individual lamina with an intraoperative hinge fracture clearly identified by the attending surgeon will be treated as a "non-responder" within the FAS for the analysis of the lamina retention rate, hinge fracture, and bone union of the gutter.

2. Rationale for the non-inferiority margin According to the CONSORT extension for noninferiority trials, the rationale for the non-inferiority margin should be stated. The current protocol uses a 20% margin based on the MCID. However, further clinical rationale for selecting "20%" would be helpful. For example, please explain if this 20% margin is consistent with previous studies, or if it is an acceptable difference considering the potential benefits of the suture-anchor technique (such as shorter operative time or lower cost).

Response: To our knowledge, there are no previous randomized non-inferiority trials comparing these specific techniques using the JOA score recovery rate. Therefore, authors determined the 20% margin through consensus, considering it to be slightly more than half of the established MCID (52.8% as described in reference #25). It is important to note that this margin was selected based solely on clinical equivalence in functional recovery; we did not lower the margin to account for potential advantages of the suture-anchor technique, such as reduced operative time or cost. We have added this explanation to the "Sample size" section.

P13: Line 248-252: The non-inferiority margin was set at 20%. As no previous non-inferiority RCTs have compared these techniques, this margin was determined by expert consensus to be approximately half of the known MCID (52.8% [25]) for the JOA recovery rate, ensuring a clinically acceptable difference regardless of potential benefits in operative time or cost.

To Editorial Office

a) If there are ethical or legal restrictions on sharing a de-identified data set, please explain them in detail (e.g., data contain potentially identifying or sensitive patient information, data are owned by a third-party organization, etc.) and who has

---

## [Decision Letter · Decision Letter 1]

18 Mar 2026

Clip-plate versus Suture-Anchor in Double-door Laminoplasty for Degenerative Cervical Myelopathy: Protocol for a Multicenter, Non-Inferiority, Randomized Controlled Trial

PONE-D-25-63840R1

Dear Dr. Yoshii,

We’re pleased to inform you that your manuscript has been judged scientifically suitable for publication and will be formally accepted for publication once it meets all outstanding technical requirements.

Kind regards,

Koji Akeda

Academic Editor

PLOS One

Additional Editor Comments (optional):

Thank you very much for submitting your highly valuable research protocol.

We sincerely appreciate your careful responses to the reviewers' comments.

We are pleased to inform you that your manuscript has been accepted for publication in PolsOne Journal. We look forward to your continued contributions in the future.

Reviewers' comments:

Reviewer's Responses to Questions

**Comments to the Author**

1. Does the manuscript provide a valid rationale for the proposed study, with clearly identified and justified research questions?

Reviewer #1: Yes

Reviewer #3: Yes

2. Is the protocol technically sound and planned in a manner that will lead to a meaningful outcome and allow testing the stated hypotheses?

Reviewer #1: Yes

Reviewer #3: Yes

3. Is the methodology feasible and described in sufficient detail to allow the work to be replicable?

Reviewer #1: Yes

Reviewer #3: Yes

4. Have the authors described where all data underlying the findings will be made available when the study is complete?

The PLOS Data policy requires authors to make all data underlying the findings described in their manuscript fully available without restriction, with rare exception, at the time of publication. The data should be provided as part of the manuscript or its supporting information, or deposited to a public repository. For example, in addition to summary statistics, the data points behind means, medians and variance measures should be available. If there are restrictions on publicly sharing data—e.g. participant privacy or use of data from a third party—those must be specified.requires authors to make all data underlying the findings described in their manuscript fully available without restriction, with rare exception, at the time of publication. The data should be provided as part of the manuscript or its supporting information, or deposited to a public repository. For example, in addition to summary statistics, the data points behind means, medians and variance measures should be available. If there are restrictions on publicly sharing data—e.g. participant privacy or use of data from a third party—those must be specified.

Reviewer #1: Yes

Reviewer #3: Yes

5. Is the manuscript presented in an intelligible fashion and written in standard English?

Reviewer #1: Yes

Reviewer #3: Yes

You may also provide optional suggestions and comments to authors that they might find helpful in planning their study.

Reviewer #1: I extend my gratitude to the authors for their sincere consideration of my opinions. This process has also updated my knowledge and proved highly valuable.

The literature on the mechanical properties of Laminaclip2 is interesting. I also gained insight into the situation by learning that it is managed as a type of plate in the authors' country. The term “clip-plate” strikes me as a good designation, clearly indicating it addresses different material properties than conventional miniplates while suggesting a plate system-like outcome.

The other revisions also seem satisfactory, reducing the potential for reader confusion.

Once again, I express my respect for the authors' efforts and wish this research every success.

Reviewer #3: The authors have adequately addressed the points raised during the previous review. The revised protocol is clear and well-prepared, and I believe no further comments are necessary from the reviewer’s perspective.

.

Reviewer #1: No

Reviewer #3: No

---

## [Editor Report · Acceptance letter]

PONE-D-25-63840R1

PLOS One

Dear Dr. Yoshii,

I'm pleased to inform you that your manuscript has been deemed suitable for publication in PLOS One. Congratulations! Your manuscript is now being handed over to our production team.

Kind regards,

on behalf of

Dr. Koji Akeda

Academic Editor

PLOS One